# Sea surface temperature modulates El Niño and La Niña driven leptospirosis patterns: Evidence from causal machine learning in Colombia

Juan David Gutiérrez[1]*, Juan Wilches-Vega[2], Fabián Galvis-Serrano[3], Holver Parada-Jurado[4], Javier Cortes-Ramírez[2]

1 Universidad de Santander, Facultad de Ciencias Médicas y de la Salud, Instituto Masira, Bucaramanga, Colombia, 2 Universidad de Santander, Facultad de Ciencias Médicas y de la Salud, Programa de Medicina, Cúcuta, Colombia, 3 Universidad de Santander, Facultad de Ciencias Médicas y de la Salud, Grupo de Investigación BIOGEN, Cúcuta, Colombia, 4 Universidad de Santander, Facultad de Ciencias Básicas, Cúcuta, Colombia

* jdgutierrez@udes.edu.co

## Abstract

Leptospirosis is a zoonotic disease prevalent in tropical regions influenced by climatic factors such as precipitation and soil moisture, which are regulated by the El Niño-Southern Oscillation (ENSO). This study examines the causal relationship between El Niño and La Niña episodes and leptospirosis cases in Colombia at the municipal level from 2007 to 2023. Using an ecological longitudinal design, we analyzed laboratory-confirmed cases from the National Public Health Surveillance System, environmental data from remote sensing, and socioeconomic data, employing a causal machine learning framework with doubly robust estimation and overlap weighting. We estimated the Average Treatment Effect (ATE) and the Conditional Average Treatment Effect (CATE) for three scenarios: Neutral vs. La Niña, Neutral vs. El Niño, and El Niño vs. La Niña. Results showed 10,629 cases, predominantly in males, with the highest incidence in Cali, Barranquilla, San José del Guaviare, and Cartagena. La Niña was associated with a 1.2 percentage point reduction in the probability of excess leptospirosis cases (ATE = -0.012, 95% CI: -0.015 – -0.008), while El Niño corresponded with a 7.2 percentage point increase in the probability of excess leptospirosis cases (ATE = 0.072, 95% CI: 0.041 – 0.103) compared to Neutral episodes. The El Niño vs. La Niña comparison showed no significant effect. As sea surface temperatures rose in the Pacific Ocean off the Colombian coast, the impact of both El Niño and La Niña episodes was observed to diminish, according to the CATE analysis. Regional variations, particularly in the Orinoco and Amazon regions, seem to drive these national trends, probably due to inverse hydro-climatic responses to ENSO. Refutation tests indicated the presence of remaining bias for the scenarios Neutral vs. El Niño and El Niño vs. La Niña. These findings highlight the complex interplay between climate and leptospirosis, underscoring the need for region-specific public health strategies to mitigate climate-driven disease risks in Colombia.

**Data availability statement:** Interested readers can find the code and dataset for reproducing the results on GitHub: https://github.com/juandavidgutier/ENSO_leptospirosis.

**Funding:** The authors received no specific funding for this work.

**Competing interests:** The authors have declared that no competing interests exist.

# 1 Introduction

Leptospirosis is a globally significant zoonotic disease caused by pathogenic spirochetes of the genus *Leptospira*, primarily transmitted through contact with water or soil contaminated by urine from infected reservoir hosts such as rodents, livestock, and wildlife [1]. The disease manifests in two distinct phases: an initial acute phase characterized by nonspecific symptoms, including high fever, severe headache, myalgia, and conjunctival suffusion [2], followed by a potentially severe immune phase that may lead to life-threatening complications such as Weil's syndrome (jaundice, renal failure, and hemorrhage) or pulmonary hemorrhage syndrome [3]. Pathogenesis involves complex interactions between host immune responses and bacterial virulence factors, with genetic predisposition playing a critical role in disease severity [4]. Diagnosis remains challenging due to symptom overlap with other tropical diseases (e.g., dengue or malaria), though serological tests (MAT, ELISA) and molecular methods (PCR) are commonly employed [1]. Early antibiotic treatment with doxycycline or penicillin reduces complications, but mortality can exceed 10% in severe cases without timely intervention [3].

Globally, leptospirosis causes an estimated >1 million cases and 60,000 deaths annually, disproportionately affecting tropical and subtropical regions due to climatic factors facilitating bacterial survival and socioeconomic conditions promoting transmission [4]. In the Americas, leptospirosis is hyperendemic, with Brazil and Colombia reporting the highest regional burden of disease; seroprevalence studies indicate infection rates of 21–29% in high-risk populations (e.g., agricultural workers and urban slum dwellers) [5]. In Colombia, previous research estimated 2,186 cases nationwide between 2015–2020, highlighting occupational exposure and inadequate sanitation infrastructure as key drivers [6]. In regional hotspots such as the Córdoba department, severe clinical presentations, including renal failure in >60% of hospitalized patients are reported, underscoring gaps in surveillance and preventive infrastructure [7].

Climatic variables, such as precipitation and soil moisture, significantly influence the epidemiology of leptospirosis. Heavy precipitation increases the incidence of leptospirosis by causing flooding, which disseminates contaminated water and enhances human exposure to the pathogen [8–10]. A study in Brazil demonstrated a positive correlation between monthly rainfall and leptospirosis cases, highlighting how precipitation drives outbreaks [11]. Similarly, research in Thailand identified rainfall as a key predictor of leptospirosis incidence, reinforcing the role of water-related dissemination [12]. Additionally, soil moisture affects the survival and transmission of *Leptospira* by creating favorable conditions for the bacteria and prolonging their viability outside a host [13]. Research in urban settlements in Brazil quantified the presence of pathogenic *Leptospira* in soil samples, finding higher concentrations in soils with greater moisture content, suggesting that moist soil acts as an environmental reservoir [14]. Furthermore, studies have shown that *Leptospira* can multiply in waterlogged soils, indicating that soil is not only a passive carrier but an active reservoir for the pathogen [15].

The El Niño-Southern Oscillation (ENSO) is a natural climate phenomenon characterized by fluctuations in sea surface temperatures in the tropical Pacific

Ocean, manifesting in three episodes: El Niño, La Niña, and Neutral. El Niño involves warmer-than-average sea surface temperatures, La Niña features cooler-than-average temperatures, and Neutral episodes occur when temperatures are near average [16]. In Colombia, these episodes significantly influence regional hydro-climate patterns, affecting precipitation, temperature, and hydrological systems with notable socioeconomic consequences. During El Niño episodes, Colombia experiences reduced precipitation and elevated temperatures, particularly in the Andean, Caribbean, and Pacific regions, leading to drought conditions that impact water supply, agriculture, and increase the risk of wildfires [17]. Conversely, La Niña episodes are associated with increased precipitation, often resulting in extreme weather episodes such as flash floods, landslides, and long-term flooding of plains, particularly in the Andean and Caribbean regions [17]. Neutral episodes generally result in less extreme climate impacts, though they still contribute to variability in precipitation and temperature, with effects on streamflow and soil moisture being less pronounced but still significant [18].

In this study, we aimed to estimate the effect of El Niño and La Niña episodes on leptospirosis incidence in Colombia at the municipal scale between 2007 and 2023. We implemented a causal machine learning approach to estimate the Average Treatment Effect (ATE), and the Conditional Average Treatment Effect (CATE) of El Niño and La Niña episodes on leptospirosis, given the sea surface temperature off the Colombian Pacific coast. This research aims to provide a data-driven analysis to inform targeted public health interventions and environmental monitoring policies, to reduce the spread of leptospirosis in Colombia and other similar tropical regions where the disease is endemic.

## 2 Methods

We conducted an ecological longitudinal study (with municipalities as observational units) to evaluate the effect of El Niño and La Niña episodes on excess leptospirosis cases in Colombia from 2007 to 2023. The ethical aspects of the study were approved by the Bioethics Board of the Universidad de Santander (Minute No. 002, February 13, 2023). Additionally, our study adheres to the Strengthening the Reporting of Observational Studies in Epidemiology (STROBE) guidelines.

### 2.1 Data collection

**2.1.1 Epidemiological data.** The National Public Health Surveillance System (SIVIGILA) provided (21st October 2024) anonymized leptospirosis surveillance data from January 2007 to December 2023, in Colombian municipalities. The dataset contains laboratory-verified leptospirosis cases, with confirmation procedures conducted at various administrative levels, including municipal facilities, departmental institutions, and the National Institute of Health. To ensure data quality, we implemented a cleaning protocol that excluded entries containing data inconsistencies, such as invalid municipality codes, incorrect occurrence dates, or implausible age values (such as ages exceeding 120 years). Daily cases were grouped by month and municipality of occurrence.

In epidemiological surveillance, the excess of cases refers to the occurrence of more disease cases than would normally be expected in a specific population, time, or place. This concept is fundamental for outbreak detection and public health monitoring, as it identifies when disease patterns deviate from established baselines [19]. Unlike simple case counts, which fail to account for underlying population demographics and disease patterns, the identification of excess cases requires standardized comparison against an expected baseline derived from a reference population.

To identify the excess cases of leptospirosis in our study, we employed the Standardized Incidence Ratio (SIR) methodology. The SIR is a widely-used epidemiological measure that compares the observed number of disease cases in a study population to the expected number of cases based on the incidence rates of a reference population, adjusted for demographic factors such as age and sex. The SIR is calculated as:

SIR = Observed cases/ Expected cases

Where the expected cases are derived by applying age-specific incidence rates from the reference population (national level) to the age structure of the study population (municipal level) [20]. This standardization process is essential because

disease incidence often varies significantly across age groups, and populations may have different age distributions that could obscure direct comparisons [21].

In our implementation, we estimated the SIR using the epitools package (version 0.5-10.1) in R software [22]. The standardization process utilized WHO standard age groups and national population data as the reference population for the analysis. The WHO age groupings provide internationally recognized demographic strata that ensure consistency and comparability across epidemiological studies [23].

Note that an SIR value exceeding 1 indicates a higher-than-expected disease occurrence in the studied municipality relative to the national average, while values below 1 suggest lower-than-expected incidence compared to the reference population's incidence. For the purposes of our causal machine learning analysis, we operationally defined excess leptospirosis cases by binarizing the SIR values. Specifically, we assigned a binary value of 1 when the SIR was greater than 1 (indicating that excess cases occurred, i.e., observed cases exceeded expected cases), and 0 otherwise (indicating that excess cases did not occur, i.e., observed cases were equal to or below expected cases).

This binary operationalization transforms the continuous SIR measure into a binary outcome variable suitable for causal inference methods, where the presence or absence of excess cases represents a meaningful public health threshold. Municipalities with an SIR value greater than 1 (i.e., excess cases = 1) experienced epidemiological conditions that resulted in more leptospirosis cases than would be expected based on national patterns.

**2.1.2 Environmental data.** We used the definitions developed by four climate agencies: the National Oceanic and Atmospheric Administration (NOAA) [24], the Meteorological Office of the Australian Government (MOAG) [25], the Tokyo Climate Center (TCC) [26], and the Colombian Institute of Hydrology, Meteorology, and Environmental Studies (IDEAM) [27]. For the NOAA, the criterion to classify a month as part of an El Niño or La Niña episode is the three-month moving average of Sea Surface Temperature (SST) in the El Niño 3–4 region being greater than or equal to +0.5 °C for El Niño, and less than or equal to -0.5 °C for La Niña. For the MOAG, the criterion for El Niño is that Southern Oscillation Index (SOI) values must remain at or below -7 for three consecutive months, and for La Niña, SOI values must remain at or above +7 for three consecutive months. For the TCC, the criterion is the continued five-month moving average SST anomaly in the El Niño 3 region being greater than or equal to +0.5 °C during at least six consecutive months for El Niño, and less than or equal to -0.5 °C during at least six consecutive months for La Niña. For the IDEAM, the criterion for El Niño is a positive SST anomaly in the El Niño 3 region equal to or greater than 1.67, and for La Niña, a negative SST anomaly in the El Niño 3 region less than -1.02.

Note that these four agencies maintain distinct definitions for episode classification, our approach prioritized methodological rigor to maintain study consistency and uphold causal inference assumptions, particularly the consistency assumption (detailed in Section 2.3 Causal inference analysis). To achieve maximum reliability, we implemented a stringent classification system that required agreement across all four climate agencies' criteria. S1 File shows for each month between 2007 and 2023, whether the month was classified as part of an El Niño, La Niña, or Neutral episode, for each climate agency, along with the corresponding consensus.

To address potential confounders in our study, we followed the rationale outlined by Gutiérrez and Tapias-Rivera (2024) [28] and Gutiérrez et al. (2024) [29]. Specifically, atmospheric and oceanic indices drive hydro-climate patterns in continental regions [30–32]. These indices can introduce confounding bias through a backdoor path due to their simultaneous association with the exposure variable (El Niño and La Niña episodes) and the outcome variable (excess leptospirosis cases) through changes in hydro-climate variables associated with leptospirosis occurrence (e.g., floods). To control for confounding bias, we utilized monthly data from nine atmospheric and oceanic indices from the NOAA for the period January 2007 to December 2023 [24] (Table 1).

We obtained monthly data for the period 2007–2023 on air temperature, soil temperature up to 7 cm, volumetric soil water up to 7 cm, runoff, and rainfall from the ERA5 dataset, with a spatial resolution of 0.10 degrees [33]. To align these variable data with the municipal boundaries, we used the raster package in R software. This process identified

**Table 1. Atmospheric and oceanic indices included as potential confounders.**

| Name | Detail |
|---|---|
| SST12 | Sea surface temperature in El Niño region 1–2 |
| SST3 | Sea surface temperature in El Niño region 3 |
| SST34 | Sea surface temperature in El Niño region 3–4 |
| SST4 | Sea surface temperature in El Niño region 4 |
| SOI | (Standardized anomalies Tahiti - Standardized anomalies Darwin) sea level pressure |
| ESOI | Standardized anomalies Indonesia sea level pressure |
| NATL | Sea surface temperature North Atlantic (5–20°North, 60–30°West) |
| SATL | Sea surface temperature South Atlantic (0–20°South, 30°West-10°East) |
| TROP | Sea surface temperature Global Tropics (10°South-10°North, 0–360) |

pixels within each polygon and subsequently calculated the average values for each environmental variable within each municipality.

**2.1.3 Socioeconomic data.** The Multidimensional Poverty Index (MPI) is a national socioeconomic index of the percentage of households with multidimensional poverty in each municipality, according to the national census of 2018. The MPI encompasses six dimensions: education access, childhood and youth conditions, work opportunities, health access, public services, and housing quality. We sourced the MPI data from the National Terridata repository [34].

To estimate population density for each municipality per year, we divided the population of each municipality by the corresponding municipality's area. Yearly population data were based on national population estimates obtained from the National Statistics Department [35].

## 2.2 Outcome and exposure variables

The exposure variable was the occurrence of ENSO episodes, for which we evaluated three scenarios. In the first scenario, we compared Neutral episodes (control) to La Niña episodes (exposure). In the second, we compared Neutral episodes (control) to El Niño episodes (exposure). Finally, in the third scenario, we compared El Niño episodes (control) to La Niña episodes (exposure).

The outcome variable was the monthly occurrence of excess leptospirosis cases during the months compared in each scenario. For example, in the scenario of Neutral vs. El Niño, the consensus of the four climate agencies indicated that the Neutral episodes were: May 2012 to May 2014, February 2017 to August 2017, May 2018 to August 2018, and August 2019 to June 2020, i.e., controls in the scenario Neutral vs El Niño. Similarly, according to the consensus of the four climate agencies, the months from July 2015 to January 2016 and August 2023 to November 2023 were the months of El Niño episodes, i.e., exposure in the scenario of Neutral vs. El Niño. We included the excess leptospirosis cases for these same months as both the control and exposure outcomes.

Note that each scenario emulates an experimental design using observational data (see section 2.4 Causal inference analysis), with its respective control and exposure groups. We evaluated the effect of the exposure variable on the excess cases of leptospirosis for each scenario.

## 2.3 Variable selection strategy

The selection of covariates for inclusion in the causal model was guided by a combination of substantive domain knowledge, temporal precedence, and causal diagram-based principles to satisfy the backdoor criterion [36]. We began by constructing a Directed Acyclic Graph (DAG) (see section 2.4 Causal inference analysis) that formalized the hypothesized causal relationships among the exposure and outcome variables, and a comprehensive set of environmental, climatic,

socioeconomic, and spatiotemporal factors. The DAG was developed through a systematic review of the epidemiological literature on leptospirosis and ENSO-driven health impacts.

All covariates included in the DAG were selected based on their role in the effect of ENSO episodes on excess leptospirosis cases. This principled, DAG-informed strategy minimizes the risk of overadjustment or underadjustment bias and ensures that the resulting causal estimates are interpretable as the total effect of ENSO episodes on excess leptospirosis cases, unconfounded by observed covariates.

## 2.4 Causal inference analysis

The goal of causal inference through the backdoor path is to emulate an experimental design using observational data, analyzing the relationship between variables and adjusting for valid confounders, to avoid introducing biases (e.g., collider bias) [36]. We represented our prior knowledge about the relationships between variables and explicitly stated our causal assumptions through the DAG [37]. We included the exposure of the three compared scenarios: Neutral vs. La Niña, Neutral vs. El Niño, and El Niño vs. La Niña, in the DAG as the exposure node. The outcome variable was the excess leptospirosis cases (Fig 1).

Note that the observed association between H-C variables and MPI, and H-C variables with population density in Colombian municipalities, as depicted in our DAG, can be historically contextualized. From the colonial era onwards, regions characterized by challenging climates were often perceived as undesirable for mainstream settlement. This led to their disproportionate inhabitation by marginalized ethnic groups, including indigenous and Afro-descendant communities [38]. Consequently, these territories, exemplified by the Amazon and Pacific regions with their high precipitation and temperature, have experienced persistent governmental neglect from the central authority in Bogotá (the national capital city), resulting in a severe lack of social development and essential provisions for their inhabitants [38,39].

Please note that the variables Municipality, Municipality-Year, and Municipality-Year-Month represent the time-invariant characteristics of the municipalities, the time-variant characteristics of the municipalities for each year, and the time-variant characteristics of the municipalities for each year and each month, respectively.

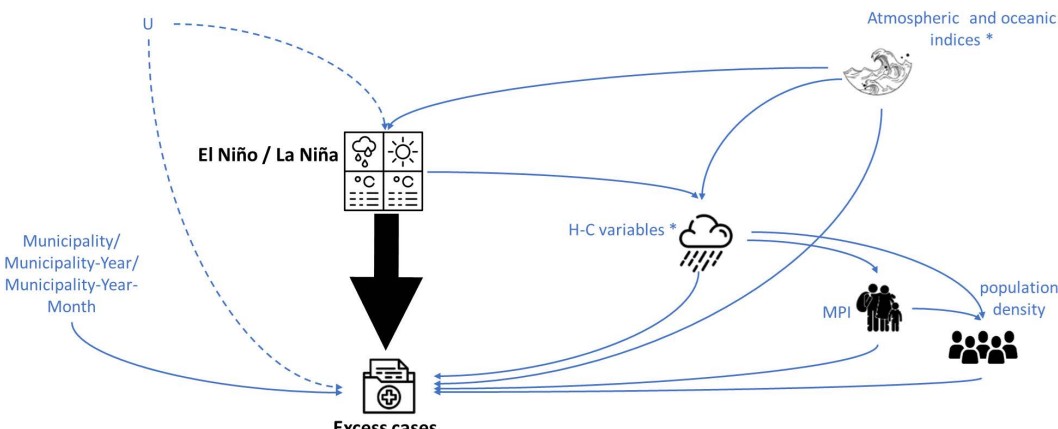

**Fig 1. Directed Acyclic Graph (DAG) with the hypothesized causal relationships between the episodes of El Niño and La Niña (exposure variable) and the excess cases of leptospirosis (outcome variable).** The black arrow represents the specific causal association of interest. The node labeled "U" represents potential unmeasured confounders, which could introduce remaining confounding bias into our estimates of the exposure variable's effect on excess leptospirosis cases. H-C variables = air temperature, soil temperature, volumetric soil water, runoff, and rainfall. MPI = Multidimensional Poverty Index. Municipality = Descriptor variable for each municipality. Municipality-Year = Descriptor variable for each municipality in each year. Municipality-Year-Month = Descriptor variable for each municipality in each year and each month. The asterisk in the nodes Atmospheric and oceanic indices and H-C variables indicates the association between the variables included in both nodes.

Note that the variables were included in the adjustment of the model to estimate the effect of ENSO episodes on excess leptospirosis cases, if they met two key criteria [40,41]:

1. They are common causes (i.e., potential confounders) of both the exposure (ENSO episode) and the outcome (excess leptospirosis cases), or

2. They are ancestors of such common causes, to prevent indirect confounding.

This approach ensured that all backdoor paths between the exposure and outcome were blocked, while avoiding the inclusion of:

- Colliders which could induce spurious associations [41],

- Mediators that were excluded from the adjustment set if they were associated with common causes, and to preserve the total causal effect of ENSO episodes [42].

We estimated the Average Treatment Effect (ATE) with a 95% confidence interval (CI) for each of the three analyzed scenarios. The ATE represents the average difference between municipalities during exposure episodes compared to control episodes. Note that our exposure and outcome variables are binary, which implies that our estimates of the ATE should be expressed in percentage points relative to the probability of excess leptospirosis cases. A positive value of the ATE indicates an increased probability of excess leptospirosis cases, whereas a negative value indicates a reduced probability of excess leptospirosis cases. Also note that the ATE calculates a single average effect across the entire study population and all observed conditions, representing the overall average effect. Furthermore, we assessed the Conditional Average Treatment Effect (CATE) on SST12, the sea surface temperature of the Pacific Ocean off the Colombian coast. The CATE indicates the heterogeneous effect of El Niño and La Niña episodes, given the sea surface temperature in the Colombian Pacific coasts (i.e., El Niño region 1–2).

Our estimates of the ATE and CATE can be interpreted as causal (i.e., what would happen to the outcome if we could manipulate the exposure) under the following assumptions [36,43–45]: 1) unconfoundedness: given all potential confounders, the outcome is independent of the exposure, 2) consistency: the observed outcome under the observed exposure value matches the counterfactual outcome under an alternative exposure value, 3) positivity: there is a nonzero probability of observing each exposure value within each combination of covariates among the studied units, 4) No measurement error: No substantial measurement errors are present such that no substantial measurement bias is induced, 5) Well specified model: The model is well-specified, such that all relevant non-linearities and statistical interactions are taken into account.

It is important to note that having multiple versions of the treatment violates the assumption of consistency [36,44–47]. To understand this, consider July 2009, according to the criteria of NOAA, MOAG, and TCC, was classified as part of an El Niño episode. In contrast, according to IDEAM, it was part of a Neutral episode. Similarly, in January 2015, NOAA and TCC classified it as an El Niño month, while MOAG and IDEAM considered it as a month with a Neutral episode. Incorporating both months (July 2009 and January 2015) in our analysis we violate the consistency assumption in our estimation of the effect of El Niño episodes on excess leptospirosis cases, therefore, we attempted to ensure the consistency assumption by avoiding multiple versions of the treatment, and for this reason, we implemented a classification system that required agreement across all four climate agencies' criteria.

## 2.5 Machine learning implementation

Before estimating the effect, and with the aim of improving precision and addressing potential violations of the positivity assumption, we applied overlap weighting [48]. Note that overlap weighting requires the estimation of the propensity score. However, due to computational restrictions in estimating the propensity score required by the overlap weighting, we

used logistic regression with Ridge regularization, incorporating 5-fold cross-validation and 30,000 iterations, implemented via scikit-learn's LogisticRegressionCV [43]. After obtaining the propensity score through logistic regression with Ridge regularization, we estimated the overlap weights ($w_i$) as follows:

$$w_i = T_i \cdot (1 - e(X_i)) + (1 - T_i) \cdot e(X_i)$$

where $T_i$ is the treatment indicator and $e(X_i)$ is the estimated propensity score for unit $i$. This weighting scheme assigns a higher weight to units in regions of covariate space with substantial overlap between treatment groups, effectively down-weighting extreme propensity scores and improving covariate balance [49,50].

Then, we estimated the causal effects of El Niño and La Niña episodes on excess leptospirosis cases, implementing a doubly robust learning framework using a sparse version of the Doubly Robust (DR) Learner algorithm from the EconML Python package (version 0.15.1) [51]. The DR-Learner combines two key components—outcome regression and propensity score modeling—in a way that ensures the treatment effect estimator remains consistent if either model is correctly specified, thereby providing robustness against model misspecification [52,53]. This approach is particularly advantageous in observational studies where unmeasured confounding may be present, as it leverages both the predictive power of machine learning algorithms and the theoretical foundations of causal inference to produce reliable treatment effect estimates [54].

Specifically, we employed a random forest regressor with 100 trees to estimate the outcome model and the propensity score, using a sparse version of the DR-Learner algorithm. This non-parametric approach allows for flexible modeling of complex non-linear relationships, without imposing restrictive functional form assumptions. Furthermore, to capture potential non-linearities and interactions among covariates, we applied polynomial feature expansion up to degree 3 using scikit-learn's PolynomialFeatures. The resulting high-dimensional feature set was then used within a sparse regression framework (Lasso-regularized regression) during the final stage of effect estimation. This allowed for automatic feature selection, reducing model complexity and enhancing interpretability, while preserving sensitivity to heterogeneous effects [55]. All models implementing DR-Learner were fitted using 5-fold cross-validation and a maximum of 30,000 iterations to ensure convergence. Random states were fixed throughout to guarantee reproducibility of results.

The dataset was partitioned, allocating 80% of the samples for training and retaining 20% for testing. The training set was used to fit the nuisance models (propensity score and outcome regression), including the calculated overlap weights, within the DR-Learner. The test set was then used to evaluate the final estimated model of the effect of ENSO episodes on excess leptospirosis cases.

To estimate the CATE of each scenario, we focused on the modifying role of SST12 on the effect of ENSO episodes on excess leptospirosis cases. After fitting the DR-Learner model, we generated a grid of 100 evenly spaced SST12 values spanning the observed range within the training data. To estimate the CATE at each point along this SST12 grid, we held the other effect modifier variables constant at their respective mean values observed in the training set. Given that the fixed effect variables (i.e., municipality, municipality-year, and municipality-year-month) required normalization for inclusion in the model due to computational requirements in Python, these were held at their mean values during prediction. This methodological approach allows us to isolate the marginal effect of SST12, illustrating how the effect of ENSO episodes on excess leptospirosis cases varies with SST12 (i.e., heterogeneous effect), while controlling for the spatio-temporal structure of the data. For each SST12 value in the grid, we predicted the local treatment effect, resulting in a continuous CATE curve. Simultaneously, we calculated the corresponding 95% CI.

As mentioned above, our exposure and outcome variables are binary, which implies that our estimates of the ATE of the effect of ENSO episodes on excess leptospirosis cases should be expressed in percentage points relative to the probability of excess leptospirosis cases (e.g., an ATE of -0.010 corresponds to a 1.0 percentage point reduction in probability).

## 2.6 Robustness tests

It is important to note that an ideal epidemiological study is one that is free from bias; that is, the estimated effect can be considered truly causal [36,45,46]. However, all observational epidemiological studies are susceptible to various types of bias, including confounding bias, information bias, misclassification bias, selection bias, inference bias, collider bias, and others [43]. In this study, we have employed several methods to address these biases, including causal machine learning techniques. Note that if our methods to address these biases were completely efficient, all types of bias would be tackled, and no signal of bias could be found in refutation tests. Nevertheless, our estimations may still be influenced by some of these biases. In this regard, we conducted a series of robustness tests to detect the presence of remaining bias in our estimates, even after implementing multiple strategies to mitigate different sources of bias.

To evaluate the robustness of our causal estimates to potential sources of bias we conducted four refutation tests using the DoWhy Python library [56]. These tests are designed to probe the validity of the estimated causal effect by deliberately introducing perturbations in the estimation. Unlike other sensitivity analyses (e.g., Rosenbaum bounds), which assess how strong an unmeasured confounder would need to be to overturn a result, refutation tests take a different approach: they ask whether the estimated effect changes significantly under conditions where no true causal effect should exist or where the estimate should remain stable.

In this framework, remaining bias refers to any systematic deviation in the estimated effect caused by different sources of bias that may invalidate the causal interpretation (i.e., an interpretation free of bias), even if the point estimate appears precise. A robust causal model should yield stable estimates under perturbations and produce null effects when applied to simulated null scenarios. The refutation tests implemented were:

1. Adding a random common cause: We introduced a simulated binary variable unrelated to the true data-generating process but artificially made to affect both the treatment and outcome. If the original estimate changes significantly, it suggests the model is sensitive to unobserved confounders.

2. Bootstrap resampling: We re-estimated the causal effect on 1,000 bootstrap samples (sampling with replacement). The p-value is computed as the proportion of bootstrap estimates that are more extreme than the original estimate under the null hypothesis of no effect.

3. Adding a randomly generated outcome: We replaced the true binary outcome (excess leptospirosis cases) with a random binary variable (Bernoulli-distributed) independent of all covariates and treatment. The expected causal effect should be close to zero; any non-zero estimate indicates model overfitting or incorrect specification.

4. Adding a placebo treatment: We replaced the actual treatment variable with a randomly generated binary variable. The estimated effect should be indistinguishable from zero if the model is correctly specified and robust.

Note that adding a random common cause and the bootstrap resampling test correspond to synthetic negative controls with an invariant transformation, whereas adding a randomly generated outcome and a placebo treatment correspond to synthetic negative controls with a nullifying transformation.

In our code, the argument num_simulations = 50 was used across all refutation tests. The argument num_simulations specifies the number of times the refutation procedure is repeated to assess the robustness of the estimated causal effect. In all the refutation tests, we use Bootstrap to approximate the frequency that the test value falls in the null distribution (i.e., the distribution of the same quantity, generated using the Bootstrap method, under specific conditions which vary from test to test), following the method suggested by Rousselet et.al. [57]. The frequency is interpreted as a probability and provides a p-value for the null hypothesis. In all tests, the p-value assesses whether the new effect estimate under perturbation is statistically different from the original estimate (or from zero, in the case of placebo and random outcome). A $p$-value $< 0.05$ indicates the presence of remaining bias in the original estimate.

Finally, during the process of estimating the effect of El Niño and La Niña episodes on excess leptospirosis cases, we transformed continuous variables into binary values using the median as threshold. We employed the R package DAGitty (version 0.3-1) [58] to evaluate the consistency between our hypothesized causal structure represented by the DAG and the dataset (i.e., conditional independence assumptions). Additionally, we utilized the R package ggdag (version 0.2.10) [59] to determine the appropriate adjustment for causal inference, and we conducted refutation tests using the DoWhy Python package (version 0.11.1) [56].

Interested readers can find the code and dataset for reproducing the results on GitHub: https://github.com/juandavid-gutier/ENSO_leptospirosis. To adhere to data protection regulations and uphold ethical research standards, all personally identifiable information was eliminated during the preprocessing phase. Please note that the shared dataset does not contain any potentially identifying participant information, because the SIVIGILA provided the data, which had been anonymized. Consequently, the information available in the GitHub repository excludes any details that could potentially identify participants. The dataset featured in the repository consists solely of aggregated data that does not allow for identification.

## Results

During the 17-year study period (2007–2023), a total of 10,627 cases of leptospirosis were reported in the 1,122 Colombian municipalities. Most cases occurred in males (74.1%, n = 7,872). The years 2010, 2011, and 2012 had the most reported cases, with 1,198, 1,188, and 1,008 cases, respectively. Over the last four years, a consistent decrease in the number of cases has been observed (Fig 2).

The top four municipalities with the highest number of cases during the study period were Cali (691), Barranquilla (608), San José del Guaviare (445), and Cartagena (427) (Fig 3a). Some of the municipalities with the highest average SIR during the study period were located in the East and South of the country, corresponding mainly to the Amazon and Orinoco regions (Fig 3b).

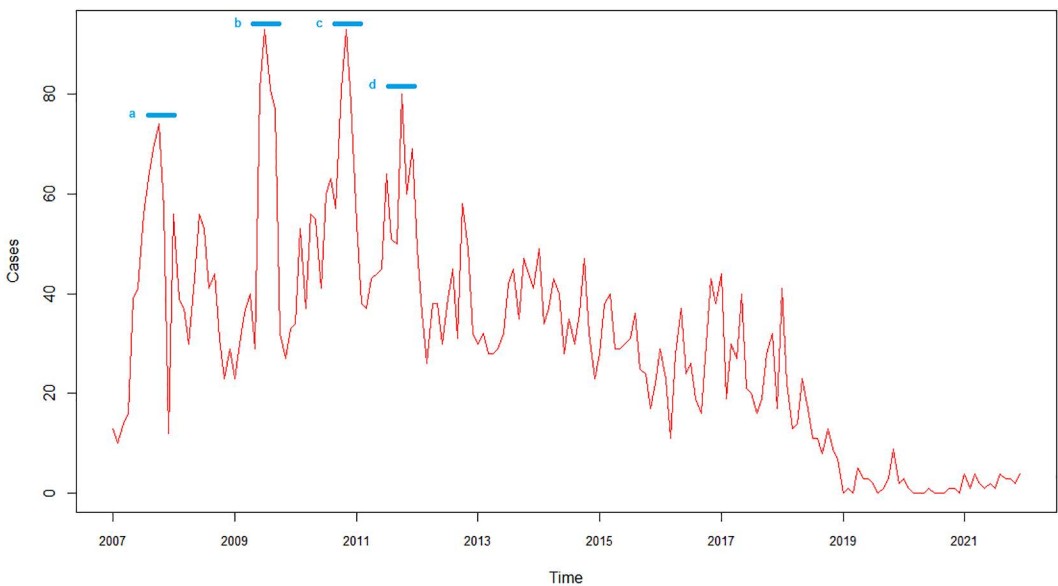

**Fig 2. Monthly time series of leptospirosis cases from 2007 to 2023 for the entire country. The blue lines indicate the months with the highest number of cases, which occurred in October 2007 (a), July 2009 (b), November 2010 (c), and October 2011 (d).**

a

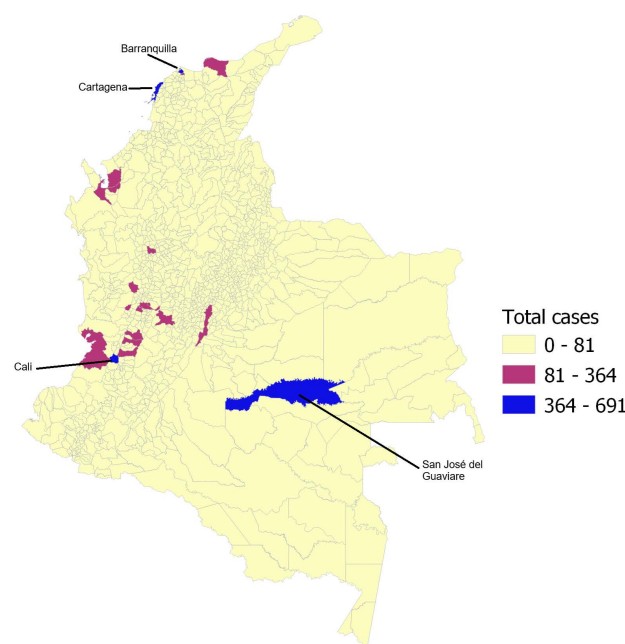

b

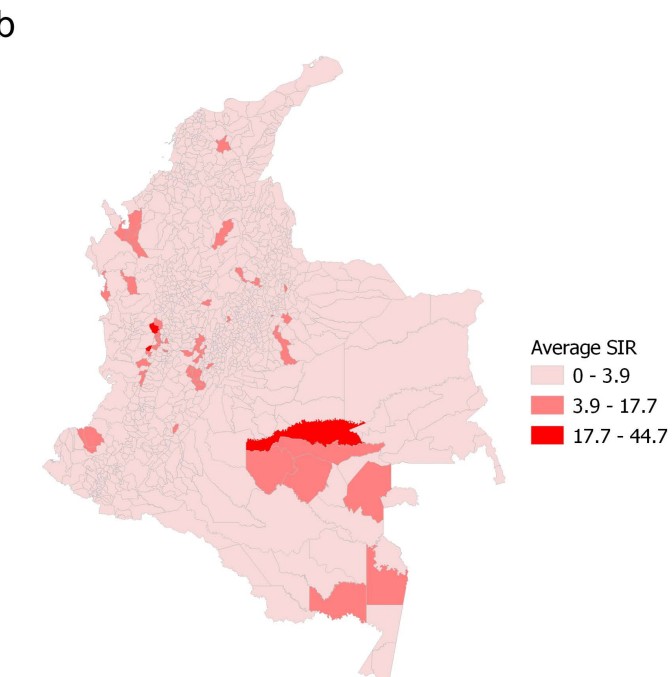

**Fig 3. Total cases per municipality for the period 2007–2023 (a), and average Standardized Incidence Ratio (SIR) of leptospirosis per municipality for the same period (b).** The municipalities with the highest number of cases are shown (a). The map was created using QGIS software, with the basemap shapefile sourced from the Colombian National Geostatistical Framework, an openly available resource (https://www.dane.gov.co/files/geoportal-provisional). The terms of use for the base shapefile are compatible with CC-BY 4.0 (https://geoportal.dane.gov.co/acerca-del-geoportal/licencia-y-condiciones-de-uso/#gsc.tab=0).

The variables air temperature, soil temperature, volumetric soil water, runoff, rainfall, MPI, and population density were identified as colliders (i.e., a variable simultaneously affected by two or more covariates) in our DAG. The inclusion of these variables in the estimation of the effect of the episodes of El Niño and La Niña on excess leptospirosis cases induces collider bias [37], and for this reason, these variables were excluded from the estimation of the ATE and CATE.

The ATE estimated for comparing Neutral and La Niña episodes was -0.012 (95% CI = -0.015 – -0.008). This indicates that the exposure to La Niña conditions was found to reduce the probability of excess leptospirosis cases by 1.2 percentage points relative to Neutral episodes. The result of the ATE for comparing Neutral and El Niño episodes indicated an increase of 7.2 percentage points in the probability of excess leptospirosis cases (ATE = 0.072, 95% CI = 0.041–0.103) during El Niño episodes compared to Neutral episodes. The scenario comparing El Niño vs La Niña episodes showed an ATE estimation not different from zero, according to the 95% CI (ATE = -0.008, 95% CI = -0.018 – 0.002) (Fig 4).

The estimation of the CATE conditioned on SST12 for the scenarios Neutral versus La Niña and Neutral versus El Niño indicated a tendency toward a lesser effect of La Niña and El Niño compared to Neutral episodes as the temperature in SST12 increased. Temperatures below 21.5 °C in SST12 increase (according to the 95% CI) the probability of excess leptospirosis cases during El Niño episodes relative to Neutral episodes, but temperatures above 25 °C reduced the probability of excess cases (Fig 5b). For the scenario for comparing Neutral vs La Niña episodes, according to the 95% CI, temperatures above 24°C reduced the probability of excess leptospirosis cases (Fig 5a). Note that the effect of the CATE conditioned on SST12 was larger for the scenario comparing Neutral versus El Niño episodes (Fig 5b). In the El Niño versus La Niña scenario, an increasing trend was observed, where the effect of La Niña compared to El Niño episodes increased as the temperature in SST12 increased. However, this effect was not significantly different from zero in most of the temperatures, according to the 95% CI (Fig 5c).

Table 2 presents the results of the four robustness tests across the three comparison scenarios. The estimated effect corresponds to the original ATE from the main analysis—that is, the average difference in the probability of excess leptospirosis cases between municipalities during ENSO episodes (e.g., El Niño) and control periods (e.g., Neutral). The new effect is the re-estimated ATE after applying each refutation test. All effects are reported in percentage points of probability.

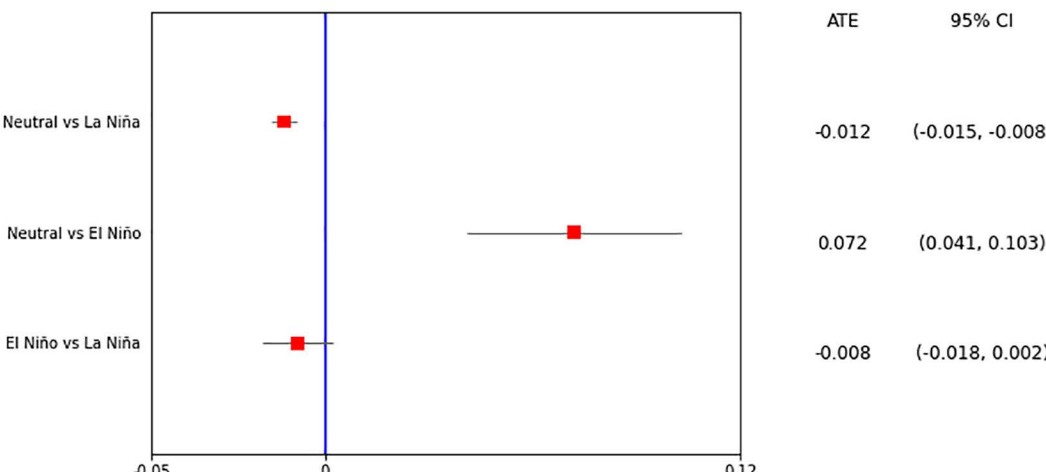

**Fig 4. Average Treatment Effect (ATE) for the different scenarios analyzed.** The x-axis represents the ATE, and the red squares indicate the point estimates of the ATE. The horizontal black lines represent the associated 95% CI. The blue line corresponds to the null effect, where the ATE is equal to zero.

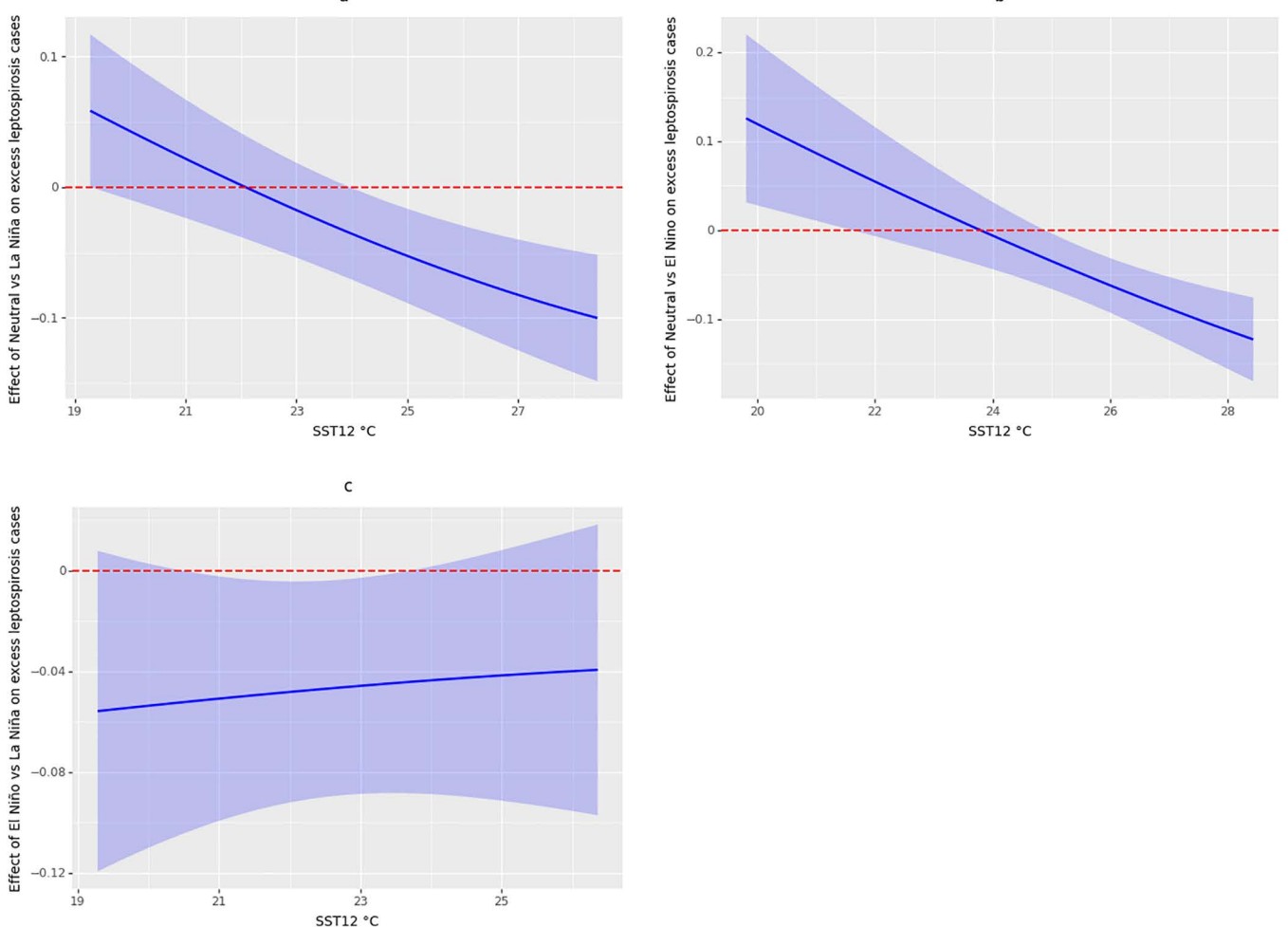

**Fig 5. Conditional Average Treatment Effect (CATE) on temperature in sea surface temperature in El Niño region 1-2 (SST12), for the three scenarios analyzed in the study: (a) Neutral vs. La Niña, (b) Neutral vs. El Niño, (c) El Niño vs. La Niña.** The blue line represents the average effect, and the blue shaded area corresponds to the 95% CI. The dotted red line represents the effect equal to zero.

The p-values in parentheses evaluate whether the refutation test induced a statistically significant change in the estimated effect. For example, in the Neutral vs. La Niña scenario, the test for the addition of a random common cause yielded a p-value of 0.32, indicating that the distribution of the added random common cause estimate does not differ significantly from the original estimate. In contrast, if a refutation test produces p-values less than 0.05 (i.e., it differs significantly from the original estimate), this suggests the presence of remaining bias.

In Table 2, for the addition of a random common cause and bootstrap resampling tests, a p-value greater than 0.05 indicates that the re-estimated effect (new effect) is not statistically different from the original ATE, suggesting that the estimate is robust to remaining bias. For the tests involving the addition of a randomly generated outcome and a placebo treatment, the expected causal effect is zero under the null hypothesis. In these cases, a new effect close to zero and a p-value greater than 0.05 indicate that the estimation procedure does not produce spurious associations in the absence of a true causal relationship, thereby supporting its validity. Note that values of exactly 0.000 in the estimated effect for the test of adding a random common cause reflect the expected null effect under the simulation.

**Table 2. Robustness tests to evaluate the estimate of the effect on excess leptospirosis cases for the three scenarios analyzed. Statistical significance is shown in parentheses (p-value); a p-value < 0.05 indicates the presence of remaining bias in the estimate; p-values < 0.05 are shown in bold.**

| Scenario | Add a random common cause (p-value) | Bootstrap resampling (p-value) | Add a randomly generated outcome (p-value) | Add a placebo treatment (p-value) |
|---|---|---|---|---|
| Neutral vs La Niña | Estimated effect = -0.012 | Estimated effect = -0.012 | Estimated effect = 0.000 | Estimated effect = -0.012 |
| | New effect = -0.012 (0.26) | New effect = -0.008 (0.08) | New effect = 0.017 (0.23) | New effect = -0.001 (0.46) |
| Neutral vs El Niño | Estimated effect = 0.072 | Estimated effect = 0.072 | Estimated effect = 0.000 | Estimated effect = 0.072 |
| | New effect = 0.071 (0.16) | New effect = 0.009 (**<0.01**) | New effect = -0.053 (0.33) | New effect = -0.019 (0.47) |
| El Niño vs La Niña | Estimated effect = -0.008 | Estimated effect = -0.008 | Estimated effect = 0.000 | Estimated effect = -0.008 |
| | New effect = -0.008 (0.28) | New effect = -0.006 (**<0.01**) | New effect = 0.005 (0.37) | New effect = -0.006 (0.42) |

## Discussion

Our results evidence a significant effect of El Niño and La Niña episodes compared with Neutral conditions. Particularly, the ATE shows a reduction in the probability of excess leptospirosis cases during La Niña and an increase in the probability of excess cases during El Niño, concerning Neutral episodes. Meanwhile, the CATE conditioned on temperature in the SST12 region displays a negative relationship between the effect of El Niño and La Niña, with the sea surface temperature in the SST12 region compared to Neutral episodes.

While leptospirosis incidence is well-established to increase with heavy precipitation and elevated soil moisture, conditions that facilitate pathogen dissemination and survival [13,60–62], our ATE estimates reveal an unexpected pattern in Colombia. Specifically, our analysis shows an average reduction in the probability of excess leptospirosis cases during La Niña episodes compared to Neutral episodes. Conversely, El Niño episodes are associated with an average increase in the probability of excess leptospirosis cases relative to Neutral conditions. This finding appears counter-intuitive given the general understanding that La Niña episodes typically bring increased precipitation and enhanced hydrological activity, particularly in the Andean, Caribbean, and Pacific regions, while El Niño episodes lead to reduced rainfall and higher temperatures in these areas [16,17].

This observed inverse relationship at the national average level could be explained by the heterogeneous hydro-climatic response of Colombia's distinct geographical regions to ENSO episodes. While the Pacific, Caribbean, and Andean regions indeed experience wetter conditions during La Niña and drier conditions during El Niño, exhibiting strong generalized synchronization with ENSO for positive (negative) hydrological anomalies [63]. The eastern and southern plains, comprising the Orinoco and Amazon regions, exhibit a different and often inverse behavior [63]. Previous research indicates that these zones show a certain insensitivity to ENSO-driven climatic variability [64], and during La Niña years, they may experience drier conditions, whereas during El Niño years, they can become more humid—a pattern opposite to that of the rest of the country. Critically, several municipalities with the highest average SIR for leptospirosis are located within these Orinoco and Amazon regions. Therefore, the overall national ATE reflects the combined, and regionally contrasting, hydrological responses to ENSO: a decrease in leptospirosis cases during La Niña might be influenced by drier conditions in the eastern and southern high-burden areas, while an increase during El Niño could be driven by more humid conditions in these same regions, leading to the observed national average.

This study represents a methodological difference in climate-health research by being the first to apply causal machine learning techniques to quantify the relationship between ENSO episodes and leptospirosis incidence. While previous studies have documented associations between SST and leptospirosis occurrence using traditional epidemiological approaches [62,65], our implementation of doubly robust estimation with overlap weighting provides more rigorous estimation by addressing confounding bias, consistency assumption, and model misspecification simultaneously. The application of causal machine learning in infectious disease epidemiology represents an emerging frontier

that offers interesting analytical capabilities for understanding complex environmental-health relationships compared to conventional statistical methods [66,67]. Furthermore, our approach of requiring consensus across four climate agencies (NOAA, MOAG, TCC, and IDEAM) for ENSO episode classification addresses the critical issue of treatment consistency that has been largely overlooked in previous climate-health studies. The estimation of both ATE and CATE provides unprecedented granularity in understanding how ENSO impacts vary across different climatic conditions, particularly sea surface temperature gradients. Our findings of paradoxical relationships—where La Niña episodes show negative effects on leptospirosis at the national level—challenge conventional understanding about the relationship between rainfall and leptospirosis, and highlight the importance of considering regional heterogeneity in climate-health associations, which have been insufficiently addressed in previous research focusing on local scales in Colombia [7].

An important aspect of our study was the strong focus on the complexity of the research question through the inclusion of multiple oceanic, hydroclimate, and socioeconomic variables. However, the refutation tests revealed evidence of remaining bias in our estimates, particularly demonstrated by the bootstrap resampling test. This finding underscores a fundamental limitation inherent to observational studies, where unmeasured bias remains a persistent threat to causal inference validity [68]. The presence of remaining bias suggests the existence of unobserved variables or mechanisms not captured in our approach, which alter the estimation of the effect of ENSO episodes on leptospirosis. While doubly robust methods provide some protection against model misspecification, they cannot fully address bias arising from unmeasured mechanisms or variables [53]. This limitation emphasizes the need for cautious interpretation of our causal effect estimates and highlights the importance of implementing sensitivity analyses to assess the robustness of findings to potential unmeasured sources of bias in epidemiological research [69,70].

A notable constraint of our findings stems from the quality of the epidemiological surveillance data used in this study. The leptospirosis case data were obtained from Colombia's national surveillance system (SIVIGILA), which operates primarily under a passive surveillance model. Passive surveillance systems, while cost-effective, are known to suffer from underreporting and delayed case detection, particularly in low-resource settings or for diseases with non-specific clinical manifestations like leptospirosis [71,72]. As a result, the true burden of the disease may be substantially underestimated, particularly during periods of low diagnostic awareness or system saturation. Moreover, reliance on laboratory-confirmed cases introduces detection bias, as only individuals who access healthcare services and undergo diagnostic testing are captured in the dataset. This approach may disproportionately exclude mild or asymptomatic cases and could skew results toward more severe manifestations that are more likely to be tested and reported [73].

Additionally, the diagnostic capacity for leptospirosis varies widely across municipalities in Colombia, especially between urban centers and remote rural or jungle regions. Limited access to diagnostic tools, trained personnel, and healthcare infrastructure in these areas contributes to differential underreporting, which can obscure true spatial patterns of disease and distort estimates of the association between climatic phenomena and disease incidence [74,75]. The presence of such differential misclassification poses a risk of introducing bias into our causal inference model. Future studies should aim to integrate syndromic surveillance or seroprevalence data [76], when available, to mitigate these limitations and enhance the representativeness and completeness of disease incidence data across all Colombian regions.

Despite employing a causal inference framework, our findings are subject to limitations inherent to complex climate-health systems. The causal relationship between ENSO episodes and leptospirosis incidence may be influenced by multiple simultaneous pathways, including hydrometeorological dynamics, host ecology, and human behavioral adaptations, which are difficult to disentangle using observational data alone [77]. Furthermore, socioeconomic factors, such as multidimensional poverty, probably modulate the impact of environmental variables on leptospirosis, and measuring how these variables modify the effect of climate on leptospirosis could be worthwhile. However, for our specific research question, incorporating multidimensional poverty into the causal estimation of the effect of El Niño and La Niña distorts the

estimation of interest and introduces collider bias. Please note that this also applies to rainfall and the other hydro-climate variables included in our study.

Furthermore, our study's findings should be interpreted in light of certain methodological restrictions inherent in its scale of analysis. First, the assumption that all locations within a given region respond homogeneously to ENSO is a potential oversimplification. Substantial evidence indicates that the hydro-climatic impacts of ENSO can be highly heterogeneous even at local levels, influenced by complex interactions with topography, land cover, and local atmospheric dynamics, which are not fully captured in a municipality-wide analysis [64]. Second, our environmental data, while useful for municipal assessment, may not capture the critical microclimatic variations that directly govern *Leptospira* survival and transmission. Environmental niches favorable for the pathogen, such as poorly drained soils, urban slums, or specific agricultural settings, create transmission hotspots that are averaged out when using large-scale climate data, potentially masking the true local risk [78].

Finally, our findings on the effect of El Niño and La Niña episodes on excess leptospirosis cases in Colombia should be interpreted with caution due to the changes in surveillance practices over the study period (2007–2023), which may have influenced case detection and reporting, potentially introducing inconsistencies in the data. For instance, variations in diagnostic capabilities across municipalities could affect the accuracy of case counts, as noted in studies on infectious disease surveillance [79]. Additionally, municipal-level public health policy variations, such as differences in sanitation infrastructure or occupational health programs, may have altered exposure risks and case reporting, distorting the observed associations [80]. Furthermore, access to healthcare services is often disrupted during extreme rainfall episodes, particularly during La Niña episodes, which are associated with flooding and landslides in Colombia's Andean, Caribbean, and Pacific regions [17]. Such disruptions can reduce case detection, potentially masking the true effect of La Niña on leptospirosis incidence, as highlighted in research on climate-driven health service interruptions [81]. These factors underscore the need for robust, standardized surveillance systems and adaptive public health strategies to mitigate the impact of climatic variability on disease monitoring. Lastly, for future works, it is recommended to analyze the impact of El Niño and La Niña episodes at both the local and regional scales using similar approaches, in order to account for regional heterogeneity in climate-health associations.

## Conclusions

This study reveals a complex and nationally counter-intuitive relationship between ENSO episodes and leptospirosis incidence in Colombia. Contrary to expectations based on localized hydrological effects, La Niña episodes were associated with an average decrease in the probability of excess leptospirosis cases, while El Niño episodes corresponded with an increase. This overarching national trend can be driven by the significant, and often inverse, hydro-climatic responses to ENSO across Colombia's diverse geographical regions. In particular, the distinct climatic behavior of high-burden areas in the Orinoco and Amazon regions appears to strongly influence the national average, overriding the more traditionally expected patterns seen elsewhere in the country.

Methodologically, this research demonstrates the power of applying a causal machine learning framework to untangle the intricate drivers of climate-sensitive infectious diseases. The implementation of doubly robust estimators and treatment consistency through a consensus of climate agencies is an approach that can be used to provide more rigorous causal estimates. The analysis of both average and conditional treatment effects offers a granular understanding, highlighting how the influence of ENSO on leptospirosis risk is not uniform but varies with other environmental factors, such as sea surface temperatures.

Despite the methodological approach, our estimations should be interpreted with caution, as evidenced by the robustness tests implemented. Future research can integrate higher-quality data, such as from active surveillance or seroprevalence studies, and incorporate finer-scale environmental and socioeconomic variables to build a more accurate understanding of the causal pathways linking climate phenomena to leptospirosis risk.

## Supporting information

**S1 File. Episodes of the ENSO cycle according to the National Oceanic and Atmospheric Administration (NOAA), the Tokyo Climate Center (TCC), the Meteorological Office of the Australian Government (MOAG), and the Institute of Hydrology, Meteorology and Environmental Studies of Colombia (IDEAM), between 2007 and 2023.** (XLSX)

## Acknowledgments

We thank the Colombian Ministry of Health for providing access to epidemiological data.

## Author contributions

**Conceptualization:** Juan David Gutiérrez.

**Data curation:** Juan David Gutiérrez, Juan Wilches-Vega, Fabián Galvis-Serrano, Holver Parada-Jurado, Javier Cortes-Ramírez.

**Formal analysis:** Juan David Gutiérrez.

**Investigation:** Juan David Gutiérrez, Juan Wilches-Vega, Fabián Galvis-Serrano, Holver Parada-Jurado, Javier Cortes-Ramírez.

**Methodology:** Juan David Gutiérrez.

**Project administration:** Juan David Gutiérrez.

**Resources:** Juan David Gutiérrez, Juan Wilches-Vega, Fabián Galvis-Serrano, Holver Parada-Jurado, Javier Cortes-Ramírez.

**Software:** Juan David Gutiérrez.

**Supervision:** Juan David Gutiérrez, Juan Wilches-Vega, Fabián Galvis-Serrano, Holver Parada-Jurado, Javier Cortes-Ramírez.

**Validation:** Juan David Gutiérrez, Juan Wilches-Vega, Fabián Galvis-Serrano, Holver Parada-Jurado, Javier Cortes-Ramírez.

**Visualization:** Juan David Gutiérrez.

**Writing – original draft:** Juan David Gutiérrez.

**Writing – review & editing:** Juan David Gutiérrez, Juan Wilches-Vega, Fabián Galvis-Serrano, Holver Parada-Jurado, Javier Cortes-Ramírez.

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
