## [Decision Letter · Decision Letter 0]

30 Aug 2025

PGPH-D-25-02250

Sea Surface Temperature Modulates El Niño and La Niña Driven Leptospirosis Patterns: Evidence from Causal Machine Learning in Colombia

Dear Dr. Gutiérrez

Thank you for submitting your manuscript to PLOS Global Public Health. After careful consideration, we feel that it has merit but does not fully meet PLOS Global Public Health’s publication criteria as it currently stands. Therefore, we invite you to submit a revised version of the manuscript that addresses the points raised during the review process.

Please do read through the careful reviews submitted. The work is important in exploring how climate impacts leptospirosis, but the details, especially in the figures, will need to be updated in order to more clearly communicate. Do make sure to respond to each of the points raised.

We look forward to receiving your revised manuscript.

Kind regards,

Megan Coffee, MD, PhD

Academic Editor

Journal Requirements:

1. Please note that PLOS Global Public Health has specific guidelines on code sharing for submissions in which author-generated code underpins the findings in the manuscript. In these cases, all author-generated code must be made available without restrictions upon publication of the work. Please review our guidelines at https://journals.plos.org/globalpublichealth/s/materials-and-software-sharing#loc-sharing-code and ensure that your code is shared in a way that follows best practice and facilitates reproducibility and reuse.

2. Please send a completed 'Competing Interests' statement, including any COIs declared by your co-authors. If you have no competing interests to declare, please state "The authors have declared that no competing interests exist". Otherwise please declare all competing interests beginning with the statement "I have read the journal's policy and the authors of this manuscript have the following competing interests:"

3. Some material included in your submission may be copyrighted. According to PLOS’s copyright policy, authors who use figures or other material (e.g., graphics, clipart, maps) from another author or copyright holder must demonstrate or obtain permission to publish this material under the Creative Commons Attribution 4.0 International (CC BY 4.0) License used by PLOS journals. Please closely review the details of PLOS’s copyright requirements here: PLOS Licenses and Copyright. If you need to request permissions from a copyright holder, you may use PLOS's Copyright Content Permission form.

Potential Copyright Issues:

Figure 3: please (a) provide a direct link to the base layer of the map (i.e., the country or region border shape) and ensure this is also included in the figure legend; and (b) provide a link to the terms of use / license information for the base layer image or shapefile. We cannot publish proprietary or copyrighted maps (e.g. Google Maps, Mapquest) and the terms of use for your map base layer must be compatible with our CC-BY 4.0 license. 

Reviewers' comments:

Reviewer's Responses to Questions

**Comments to the Author**

1. Does this manuscript meet PLOS Global Public Health’s publication criteria?

Reviewer #1: Partly

Reviewer #2: Partly

2. Has the statistical analysis been performed appropriately and rigorously?

Reviewer #1: Yes

Reviewer #2: No

3. Have the authors made all data underlying the findings in their manuscript fully available (please refer to the Data Availability Statement at the start of the manuscript PDF file)?

Reviewer #1: Yes

Reviewer #2: Yes

4. Is the manuscript presented in an intelligible fashion and written in standard English?

Reviewer #1: Yes

Reviewer #2: Yes

Reviewer #1: The article “Sea Surface Temperature Modulates El Niño and La Niña Driven Leptospirosis Patterns: Evidence from Causal Machine Learning in Colombia” examines the relationship between El Niño and La Niña episodes and leptospirosis cases in Colombia at the municipal level from 2007 to 2023 using a Casual Machine Learning approach.

The data and methodology used are appropriate and correctly applied, and the originality of the work arrives from the inverse of what is already accepted in the scientific community. In general, leptospirosis epidemics are driven by humid events, such as those caused by La Niña periods in the study region, and the opposite occurs during drier periods, but this work shows the contrary. Also, the novelty of the work is that it used real case records over an extended period in a region where there was no published evidence of the relationship between climate and this disease.

In order to be acepted, I consider that the quality of figures and table should be improved. All of them present low definition and contrast of colours.

Specifically, Figure 2 could show higher detail of the years of data at the x-axis, as well remark the years with more important outbreaks. In the caption, clarify if the data correspond to the whole country or some specific municipalities.

Additionally, in Figure 3, please indicate the names of the municipalities with the highest number of cases, to correspond with the text of the article.

On other hand, table 2 would be more legible if it appeared on a single page, perhaps with a smaller font size. Please consider redesigning it.

You must say which works of literature you are referring to on line 456.

For future work, it is recommended to analyze the impact of ENSO phenomena at local or regional scale with similar approaches, to take into account regional heterogeneity in climate-health associations.

Reviewer #2: This manuscript investigates the causal effects of El Niño and La Niña climatic events on leptospirosis incidence in Colombia using an ecological longitudinal design and a causal machine learning framework with doubly robust estimation and overlap weighting. The study spans data from 2007 to 2023, integrating epidemiological surveillance records, satellite-derived climate data, and socioeconomic indicators to estimate both Average Treatment Effects (ATE) and Conditional Average Treatment Effects (CATE).

- Critical methodological components are not well explained or justified, particularly regarding treatment definition, model assumptions, and how excess cases are calculated.

- Presentation of figures and tables is inconsistent or unclear, with some results hard to interpret or misaligned with the stated goals (e.g., monthly trends shown as annual aggregates).

- Key robustness and sensitivity tests are underdeveloped or raise questions about statistical validity, particularly Table 2 which appears internally inconsistent.

- The causal claims, especially the interpretation of CATE and refutation tests, are not sufficiently supported or explained, leading to ambiguity around the main findings.

Major Comments:

- Figure 2-is labeled as "Monthly time series of leptospirosis cases", yet the actual visualization appears to aggregate data annually. If monthly variation is central to your modeling (e.g., to match ENSO phases), the data resolution in the figure should reflect that. Otherwise, the figure is misleading and inconsistent with the analysis frame.

- It remains unclear how ENSO exposure is temporally aligned with disease cases. Are cases lagged behind SST anomalies? If so, what lag window was used? Causal claims hinge on this temporal structure and must be justified more rigorously.

- The paper references use of doubly robust estimation and overlap weighting but provides insufficient detail on the implementation. Which model was used for the outcome regression? What was the propensity model? Were these trained separately or in a joint framework?

- More importantly, how is “excess leptospirosis” defined for the treatment effect estimation? The phrase is repeatedly used, but the operational definition (e.g., above mean? above baseline trend? z-score threshold?) is not formally described. This is critical for reproducibility and interpretability.

- This table is particularly confusing. For example, some rows show "estimated effect = -0.010" with no explanation of units or interpretation. Why do some entries show 0.000 exactly? Is this due to rounding, or lack of effect? Also, the statistical significance labeling (p-values in bold) is introduced without a clear link to model diagnostics. What are these p-values testing—significance of the ATE or a test for hidden confounding (e.g., placebo/refutation analysis)? Please clarify.

- Additionally, the meaning of “latent bias” is vague. Are you conducting a sensitivity analysis (e.g., Rosenbaum bounds)? Or are these refutation tests based on synthetic controls or negative outcome models? The term “refutation” needs to be grounded in methodology.

- The paper claims that "as sea surface temperatures rise, the impact of ENSO diminishes." This is an interesting but highly speculative claim unless directly supported by CATE visualization or conditional models that include interaction terms. If you used sea surface temperature as a moderator, please show actual conditional effect plots. Without this, the claim risks overstating the evidence.

- Also, if SST is itself driving the definition of El Niño/La Niña, how is it being separately used as a conditioning variable in CATE? This may introduce post-treatment bias if not handled carefully.

- How are El Niño and La Niña episodes operationalized? Are you using NOAA’s ONI thresholds? Were partial years discarded? How are ambiguous or transitional periods handled? This is fundamental to the treatment/control assignment, and currently left unspecified.

Additional Concerns:

- Figure labeling and terminology: Please ensure consistency in terminology—"ENSO events" vs. "ENSO phases", "excess cases" vs. "probability of excess cases", etc.

- Units in ATE: An ATE of 0.072—does this refer to a probability? A proportion of municipalities? Cases per 100,000 population? Clarity here is essential.

- Geographical specificity: While Cali, Barranquilla, and San José del Guaviare are mentioned as high-incidence areas, no maps or spatial analysis are provided. A heatmap or spatial clustering figure would greatly improve interpretability.

Questions to Address in Rebuttal:

1. Can you clarify and visualize the actual monthly time series (as claimed) in Figure 2? Or revise the caption to accurately reflect what is shown?

2. What precisely is meant by “excess leptospirosis cases,” and how are they derived for ATE/CATE analysis?

3. Please provide methodological transparency on:

- Propensity and outcome model structure

- Variable selection strategy

- How SST was used in CATE estimation

4. Can you reconcile the internal inconsistencies in Table 2? What are the units of the estimates? What do the p-values correspond to?

5. What is the source and formal method for defining El Niño and La Niña phases? How were transitions handled?

**Do you want your identity to be public for this peer review?** For information about this choice, including consent withdrawal, please see our Privacy Policy

Reviewer #1: No

Reviewer #2: **Yes: ** Teerapong Panboonyuen

---

## [Decision Letter · Decision Letter 1]

26 Oct 2025

Sea Surface Temperature Modulates El Niño and La Niña Driven Leptospirosis Patterns: Evidence from Causal Machine Learning in Colombia

PGPH-D-25-02250R1

Dear Dr Juan David Gutiérrez

We are pleased to inform you that your manuscript 'Sea Surface Temperature Modulates El Niño and La Niña Driven Leptospirosis Patterns: Evidence from Causal Machine Learning in Colombia' has been provisionally accepted for publication in PLOS Global Public Health.

Before your manuscript can be formally accepted you will need to complete some formatting changes, which you will receive in a follow up email. A member of our team will be in touch with a set of requests. A reviewer has requested improved figure quality which may also be raised in the formatting review.

Best regards,

Megan Coffee, MD, PhD

Academic Editor

Reviewer Comments (if any, and for reference):

Reviewer's Responses to Questions

**Comments to the Author**

Reviewer #1: All comments have been addressed

Reviewer #2: All comments have been addressed

publication criteria?

Reviewer #1: Partly

Reviewer #2: Partly

3. Has the statistical analysis been performed appropriately and rigorously?

Reviewer #1: N/A

Reviewer #2: No

4. Have the authors made all data underlying the findings in their manuscript fully available (please refer to the Data Availability Statement at the start of the manuscript PDF file)?

Reviewer #1: Yes

Reviewer #2: No

5. Is the manuscript presented in an intelligible fashion and written in standard English?

Reviewer #1: Yes

Reviewer #2: Yes

Reviewer #1: The authors clearly respond to the reviewers' comments, greatly improving the work.

Figures that appear separately, not in the body of the manuscript, appear to be of poor quality.

Reviewer #2: ### Expected Revisions

To make this paper suitable for publication, the following revisions are essential:

1. Provide a transparent and replicable account of the causal modeling procedure, including data preprocessing, model architectures, feature selection, hyperparameter settings, and code availability.

2. Explicitly define how “excess leptospirosis cases” are computed and justify the chosen operationalization statistically.

3. Include analyses accounting for spatial dependence (e.g., Moran’s I tests, spatial lag models) and explore whether effects persist under temporal subsampling.

4. Conduct formal sensitivity checks (e.g., Rosenbaum bounds, placebo outcomes) to assess the robustness of causal claims under potential unmeasured confounding.

5. Strengthen the discussion to connect findings to climate adaptation and vector-borne disease control policies in Colombia and comparable tropical regions.

6. Add detailed statistical reporting (standard errors, trial counts, resampling strategy) to enhance interpretability and reproducibility.

### Questions to Address in the Rebuttal

1. What specific covariates were included in the doubly robust estimation, and how was covariate balance assessed post-weighting?

2. How do the authors ensure temporal independence across ENSO cycles, given potential lagged or cumulative climatic effects?

3. Did the authors test for nonlinearity in the SST–ENSO–leptospirosis interaction (e.g., using spline terms or kernel methods)?

4. How robust are the results to different spatial resolutions or exclusion of major urban centers such as Cali and Barranquilla?

5. Could the authors provide a supplementary table or visualization showing regional ATE/CATE distributions to clarify spatial heterogeneity?

**Do you want your identity to be public for this peer review?** For information about this choice, including consent withdrawal, please see our Privacy Policy

Reviewer #1: No

Reviewer #2: **Yes: ** Teerapong Panboonyuen
